# Monocytes in Tumorigenesis and Tumor Immunotherapy

**DOI:** 10.3390/cells12131673

**Published:** 2023-06-21

**Authors:** Xiaodie Chen, Yunqing Li, Houjun Xia, Youhai H. Chen

**Affiliations:** Center for Cancer Immunology, Faculty of Pharmaceutical Sciences, Shenzhen Institute of Advanced Technology, Chinese Academy of Sciences (CAS), Shenzhen 518000, China; xd.chen@siat.ac.cn (X.C.); yq.li@siat.ac.cn (Y.L.)

**Keywords:** monocytes, tumorigenesis, tumor microenvironment, myeloid-derived suppressor cells, tumor associated macrophages, immunotherapy

## Abstract

Monocytes are highly plastic innate immune cells that display significant heterogeneity during homeostasis, inflammation, and tumorigenesis. Tumor-induced systemic and local microenvironmental changes influence the phenotype, differentiation, and distribution of monocytes. Meanwhile, monocytes and their related cell subsets perform an important regulatory role in the development of many cancers by affecting tumor growth or metastasis. Thanks to recent advances in single-cell technologies, the nature of monocyte heterogeneity and subset-specific functions have become increasingly clear, making it possible to systematically analyze subset-specific roles of monocytes in tumorigenesis. In this review, we discuss recent discoveries related to monocytes and tumorigenesis, and new strategies for tumor biomarker identification and anti-tumor immunotherapy.

## 1. Introduction

Monocytes are short-lived mononuclear phagocytes that circulate in the blood and efficiently extravasate into tissues. Under normal physiological conditions, monocytes originate from common myeloid progenitors (CMPs) in bone marrow, where CMPs further develop into granulocyte and monocyte precursors (GMPs), then monocyte and dendritic cell precursors (MDPs). After a series of developmental processes, MDPs give rise to common monocyte progenitors (cMoPs) that subsequently turn into classical monocytes [1,2,3]. Mature monocytes are a heterogeneous population of immune cells which can be divided into two categories in mice based on surface markers, namely, Ly6C^hi^CX3CR1^low^CCR2^+^ (classical) and Ly6C^low^CX3CR1^hi^CCR2^−^ (non-classical) cells, and three categories in humans, including CD14^hi^CD16^low/−^ (classical or inflammatory), CD14^low^CD16^hi^ (non-classical or patrolling), and CD14^hi/mid^CD16^+^ (intermediate) [1,4].

Tumorigenesis can result from genetic mutations along with uncontrolled tumor cell growth and immune tolerance. Monocytes and their derived cells, such as myeloid-derived suppressor cells (MDSCs) and tumor-associated macrophages (TAMs), are frequently found in the tumor microenvironment (TME) and are involved in tumor development, angiogenesis, metastatic spread, chemotherapy resistance, and immune suppression. However, the biological functions and clinical implications of different monocyte subsets are far from fully elucidated. Recently, advances in single-cell sequencing, cytometry by time of flight (Cytof), and other techniques have helped identify previously unrecognized or indistinguishable subsets of monocytes. Their fates and specific functions in TME are being elucidated, which significantly enrich our understanding of how monocytes and their subsets directly affect the biological processes of tumorigenesis.

In this review, we first focus on the phenotypical and functional remodeling of monocytes by tumor cells. We then discuss the roles of monocytes and monocyte-derived cells in regulating tumor growth and metastasis. Finally, we discuss new cancer immunotherapeutic strategies targeting monocytes.

## 2. The Global Effects of TME on Monocytes

Monocytes originate from bone marrow and migrate to inflammatory sites to perform their functions, where they can locally differentiate into macrophages or dendritic cells. In general, monocytes and monocyte-derived cells serve three main roles in the immune system, which are phagocytosis, antigen presentation, and cytokine production. In defense, tumor cells remodel them in TME in favor of tumourigenesis by affecting their numbers, phenotype, differentiation, and function.

### 2.1. The Effect of TME on the Population and Phenotype of Monocytes

Inflammation is believed as one of the inevitable consequences during tumorigenesis, leading to the recruitment of inflammatory cells, such as monocytes, to tumor sites via the bloodstream. This explains the significant increases in the number and proportion of monocytes in cancer patients. For example, a higher proportion of classical and intermediate monocytes in the peripheral blood of non-small cell lung carcinoma (NSCLC) patients is observed and compared with those of healthy donors. The counts of monocytes are especially high in patients with histories of smoking, drinking, and liver metastasis [5]. A higher count of CD163-expressing intermediate monocytes is seen in breast cancer patients compared with healthy women [6]. This change seems cancer-type-dependent as the level of intermediate monocytes is significantly lower in squamous cell carcinoma of the head and neck (SCCHN) than that in healthy donors [7]. Similarly, a significant expansion of non-classical monocytes is observed in endometrial and breast cancer patients when compared with healthy controls [8], while a marked decrease in monocytes was detected in both cholangiocarcinoma (CCA) and hepatocellular carcinoma (HCC) patients before and after surgical procedures [9]. 

Given the fact that monocyte counts vary across different cancer types, the proportion of monocytes in peripheral blood that correlate with inflammation status may serve to be a prognostic indicator of cancer prognosis. A study highlights the role of peripheral blood absolute monocyte count (AMC) as a potential prognostic marker. In the study, the AMC level of multiple myeloma goes beyond the defined range, indicating inferior overall survival in patients [10]. Signal regulatory protein α (SIRPα) is a receptor-like transmembrane protein that suppresses both macrophage phagocytic function and inflammatory signaling. The SIRPα-CD47 interaction has been identified as a “self” signal for normal cells to avoid auto-attack by phagocytes. Notably, tumor cells have been shown to protect themselves by expressing high levels of CD47 [11]. Furthermore, more in-depth clinical research reports that increased numbers of CD14^+^SIRPα^hi^ monocytes are associated with inferior survival of follicular lymphoma, whilst an increased number of CD14^−^SIRPα^low^ subsets is correlated with better survival [12]. Changes in the number of monocytes can be used to predict patients’ responsiveness to anti-tumor therapy before surgery. This is because more CD163-expressing monocytes expand and are recruited into tumor sites after neoadjuvant chemotherapy (NAC). A significantly lower amount of CD14^low^CD16^hi^HLA-DR^+^ non-classical monocytes in patients is commonly associated with no noticeable clinical responses to NAC [6]. 

Phenotypes of monocytes are shaped diversely within TME. Currently, next-generation sequencing (NGS) has aided in relating alterations in the transcriptomic landscape of cancer-related monocytes to their phenotypic characteristics. It is noted that upregulated regulators of inflammation and monocyte migration may be correlated with elevated expression of key factors, such as immune regulatory receptors, pro-apoptotic molecules, and pro-angiogenic factors [6,8]. Serum levels of monocyte-derived cytokines, such as interleukin (IL)-6, granulocyte colony-stimulating factor (G-CSF), and granulocyte–macrophage colony-stimulating factor (GM-CSF), are also increased in lung cancer patients, particularly, G-CSF levels are positively correlated with lung cancer severity [13]. Circulating monocytes from pancreatic cancer patients show constitutive phosphorylation of signal transducer and activator of transcription (STAT) family members and impaired response upon stimulation, indicating aberrant activation and immune suppression [14]. The expression levels of those partners in monocytes or monocyte-derived cells aforementioned may be utilized as indicators for tumor grading and prognosis [15]. 

Effects of tumor cells within TME on the phenotype of monocytes have been further verified ex vivo. When co-culturing patient-derived monocytes with cancer cells (MIA PaCa-2 and HPAF-II), they displayed downregulated expression of the activation marker CD86 of M1 macrophages, suggesting compromised anti-tumor features [16]. The expression level of a cancer stemness-promoting factor CD51 [17] was increased in monocytes when co-cultured with SCCHN tumor cells [7]. Conclusively, TME profoundly influences the population and phenotype of monocytes and drives monocytes to develop into the immunosuppressive phenotype within the tumor milieu.

### 2.2. The Effect of TME on the Differentiation of Monocytes

Factors of TME have impacts on the differentiation and ultimate fate of monocytes (Figure 1). Upon stimulation with cytokines, monocytes can differentiate into either dendritic cells (DCs) or macrophages. In most cases, DCs display anti-tumor effects since they present tumor-associated antigens (TAA) and elicit cytotoxic CD8^+^ T-dependent responses. However, macrophages compete with DCs to degrade the TAA, which prevents the initiation of antigen presentation and induces immune tolerance. Apart from TAA, some tumoral factors affect the balance of monocytes in differentiating into monocyte-derived DCs or monocyte-derived macrophages. Retinoic acid (RA) within TME is found to drive intra-tumoral monocytes differentiated toward TAMs but shifted away from differentiating into DCs via suppression of DC-promoting transcription factor interferon regulatory factor-4 (IRF4) [18]. A similar phenomenon is observed in a human melanoma model in vitro. The supernatant of melanoma cell culture contains a high level of IL-10 which impedes monocyte-to-DC differentiation leading to differentiation into CD163^+^PD-L1^+^ M2-like macrophages [19]. However, it remains unclear within solid tumors how monocytes preferentially differentiate into immunosuppressive TAMs rather than immunostimulatory DCs.

Monocytes account for the major source of TAMs with M2-like phenotype in advanced tumors. In response to different environmental signals, macrophages are polarized toward two different subtypes: classically activated macrophages (M1) or alternatively activated macrophages (M2). M1-like macrophages are considered anti-tumor cells with the secretion of inflammatory factors, while M2-like macrophages are immunosuppressive with the association of initiation, progression, metastasis, and immune evasion of tumors [20]. Interestingly, several mechanisms have been elucidated that tumor-derived factors trigger monocyte differentiation into M2-like macrophages. In high-grade serous ovarian carcinoma (HGSOC), a high level of transforming growth factor alpha (TGFα) was directly related to the modulation of differentiation of monocytes into M2-like macrophages and, thereby, tumor transformation [21]. Hyaluronic acid (HA), a common component found in various tumor-associated extracellular matrices (ECM) [22], was found to contribute to the development of pro-tumor, immunosuppressive M2-like monocytes/macrophages. This is achieved by combining the effects of CD44 (HA receptor [23]) of THP-1 human monocytes and STAT3 [24,25]. In addition, the expression of intercellular adhesion molecule (ICAM)-1 and vascular cell adhesion molecule (VCAM)-1 in glioblastoma cells are enhanced by IL-1β stimulation. As a result, the interaction between monocytes and glioblastoma cells and regulated tumor-associated monocyte/macrophage polarization are enhanced [26]. Soluble IL-6 receptor (IL-6sR) and JAK-STAT signaling pathway have been found to promote differentiation of human monocytes into the CD14^+^CD163^+^ and CD206^+^ TAMs, respectively, when monocytes are co-cultured with ovarian cancer SKOV3 cells, and the differentiated TAMs acquire the ability to promote SKOV3 cell proliferation and invasion [27].

Emerging evidence suggests that TAMs are mixed cell populations with differential expression of both M1 and M2 markers, especially at an early stage of cancer. For example, both classical tissue monocytes and TAMs co-express M1/M2 markers, T cell co-inhibitory, and co-stimulatory receptors at an early stage of human lung cancer [28]. Hence, by temporally and spatially conditioning monocytes during their differentiation, monocyte-derived M1-like TAMs have the potential to reprogram to M2-like TAMs under extreme tumor conditions. In fact, tumor-derived lactic acid induces M2-like polarization of TAMs in the presence of hypoxia-inducible factor 1α (HIF-1α) [29] and histone lactylation [30]. In addition, high acidification of TME, which may be caused by lactic acid accumulation, induces G protein–coupled receptor (GPCR)-dependent expression of the transcriptional repressor ICER in TAMs. As a result, it leads to the polarization of TAMs to M2-like phenotype, promoting tumor growth [31]. 

### 2.3. The Effect of TME on the Fate of Monocytic MDSCs

MDSCs represent a group of pathologically activated neutrophils and monocytes with immunosuppressive activity [32] which include granulocytic or polymorphonuclear MDSCs (G-MDSCs or PMN-MDSCs) and monocytic MDSCs (M-MDSCs). M-MDSCs are identified as CD11b^+^Ly6G^−^Ly6C^hi^ in mice and CD11b^+^CD14^+^HLA-DR^−/lo^CD15^−^ in humans [33]. Furthermore, major histocompatibility complex class II (MHC-II) is widely used for the identification of M-MDSCs from monocytes [34]. With respect to tumorigenesis, prolonged presence of myeloid growth factors and inflammatory signals, such as GM-CSF, IL-6, and IL-1β, persistently trigger pathological activation of monocytes, which leads to the development and expansion of M-MDSC [35]. 

Several molecules that regulate the development of MDSCs have been identified, including c-Rel [36], STAT3 [37], S100A8/9 [38], TIPE2 [39], and IRF8 [37]. Our group has demonstrated c-Rel as a master effector of M-MDSC biology, which drives the expansion and immunosuppressive activity of M-MDSC [40]. As a member of the NF-κB transcription factor family [41], c-Rel is capable of promoting the transcription of immunosuppressive enzymes and other M-MDSC signature genes by directing the formation of c-Rel-C/EBPβ-pSTAT3-p65 enhanceosome [36]. Recently, we have defined a subset of M-MDSC in both mice and human melanomas that are programmed by c-Rel enhanceosome, namely c-Rel-dependent monocytes (rMos). These c-Rel^+^IL-1β^hi^Arg1^−^ rMos promote tumor growth by suppressing T cell function and maintaining a suppressive TME through IL-1β-CCL2 crosstalk [42].

## 3. Monocytes and Monocyte-Derived Cells in Tumorigenesis

Monocytes and monocyte-derived cells that reside in TME exert dual effects by either promoting or suppressing tumor growth (Table 1). Their heterogeneity in TME may explain their functional diversity. Here, we focus on the connection between monocytes and their derived cells during tumor development. Monocytes and their derived cells with different functions listed in Table 1 are further discussed in Section 3.1 and Section 3.2 below. With the help of single-cell technologies, we have been able to learn more about the functions of monocytes and their derived cells. However, most of the studies were performed using a limited number of cancer types. Whether the newly discovered monocyte subsets and their derived cells are universal across different cancer types in terms of phenotypes, signaling mechanisms, and functionality warrants further exploration.

### 3.1. Monocyte-Derived TAMs

There are a myriad of immunosuppressive macrophages within TME that are responsible for cancer progression, metastasis, and resistance to immune checkpoint therapy [51,52,53]. In most cancer models, TAMs stem from infiltrating monocytes within tumor tissues [54,55]. For example, SIGLEC1^+^ TAMs are highly enriched in aggressive breast cancer subtypes and associated with shorter disease-specific survival [8]. CCL8 secreted by SIGLEC1^+^ TAMs increases the infiltration of monocytes and the generation of more pro-tumoral TAMs. Cancer cells and TAMs secrete high levels of TNF-α that further support CCL8 production in TME. Furthermore, CCL8 stimulates breast cancer cells to produce CSF1 to support the survival and proliferation of macrophages [8,56]. In addition, TAMs also facilitate cancer progression by recruiting suppressive immune cells. Triggering receptor expressed on myeloid cells 2 (TREM2), an anti-inflammatory receptor [57], is predominantly expressed by a macrophage subpopulation residing in HCC tissues [46]. Through investigation of single-cell transcriptomes of human HCC tissues, TREM2^+^ TAMs mainly originating from S100A8^+^ monocytes are identified. This subset recruits suppressive Tregs and MDSCs to facilitate immunosuppressive TME [46].

Macrophages are abundant during bone metastasis of breast cancer in humans and mice. CD204^hi^IL-4R^hi^ macrophages are clarified as bone metastasis–associated macrophages, most of which are derived from Ly6C^+^CCR2^+^ classical monocytes recruited by CCR2. CD204^hi^IL-4R^hi^ macrophages promote bone metastasis of breast cancer in an IL-4R-dependent manner [58]. A study of NSCLC patient specimens identifies a pro-metastatic PLCG2-high-expressing subpopulation in addition to the CD14^+^CD16^+^CD81^+^ITGAX^+^CSF1R^+^ monocyte/macrophage subpopulation. CD14^+^CD16^+^CD81^+^ITGAX^+^CSF1R^+^ monocytes/macrophages secrete specific pro-fibrotic, pro-metastatic growth factors, including fibronectin 1 (FN1), cathepsins (CTSB and CTSD), and osteopontin (SPP1) [47].

Although most TAMs promote tumor growth in TME, a small number of macrophages with anti-tumor effects have also been reported. Based on the analysis of single-cell trajectory data, pro-inflammatory F4/80^+^MHCII^+^Ly6C^low^ monocyte-derived macrophages were identified to be associated with therapeutic response to avelumab [44]. These macrophages are the primary source of interferon-inducible chemokine CXCL9 [59], which attracts protective CXCR3^+^ T cells. Baseline levels of CXCL9 in patients treated with avelumab are correlated with clinical response and overall survival [44]. With the assistance of single cell pseudo-time analysis, increasing TAM subtypes with vital functions have been identified within TME. However, more experimental research is needed to define the precise roles of these subpopulations of monocyte-derived TAMs.

### 3.2. Monocytic MDSCs

MDSCs perform an important role in tumorigenesis and progression of cancer by primarily inhibiting T cell proliferation and activation via direct interaction, metabolic depletion, and cytokine modulation [35,60,61]. For instance, M-MDSCs significantly promote the activation of dormant micro-metastatic cells compared to PMN-MDSCs [62,63]. In acute myeloid leukemia (AML), elevated M-MDSC count is significantly associated with poor prognosis [64]. Circulating CD14^+^HLA-DR^lo/−^ monocytic MDSCs as an immune suppressive subset have potential clinical relevance for epithelial ovarian cancer progression [50] and HCC patients treated with trans-arterial radioembolization [65]. M-MDSCs are negatively associated with therapeutic response to immunotherapy, particularly CAR-T therapy [66,67,68,69], and with chemotherapy resistance [66,70]. 

M-MDSCs acquire immunosuppressive phenotypes in TME through multiple mechanisms (Figure 2). Cell adhesion signatures, such as integrin β1 and dipeptidyl peptidase-4 (DPP-4), which are expressed at high levels in murine and human M-MDSCs, act as regulators for tumor-promoting functions of M-MDSCs in glioblastoma [71]. Leukocyte immunoglobulin-like receptor subfamily B member 4 (LILRB4) orchestrates polarization of M-MDSCs [72,73] and suppresses the secretion of miR-1 family miRNAs, facilitating tumor migration and invasion [73]. Soluble heat shock protein 90α (HSP90α) is able to convert human monocytes into immunosuppressive M-MDSC via TLR4 signaling, which stimulates PD-L1 expression on M-MDSCs [74]. Inactivation of the type I interferon (IFN-I) pathway is required for MDSCs in cancer to acquire immunosuppressive activity while stabilizing Interferon Alpha/Beta Receptor 1(IFNAR1) combined with interferon induction therapy elicits a robust anti-tumor effect [75]. Deubiquitinating and stabilizing p65 with ubiquitin-specific peptidase 12 (USP12) impair infiltration and suppressive function of M-MDSCs, thus increasing CD8^+^ T-cell response and decelerating tumor growth [76].

A low level of MHC-II stands as a significant distinction of M-MDSC compared to monocytes. Elevated autophagy in M-MDSCs promotes the degradation of MHC-II molecules presented on the surface and impairs the anti-tumor responses. Autophagy-deficient M-MDSCs exhibit aberrant lysosomal degradation, thereby enhancing surface expression of MHC-II molecules which results in sufficient activation of tumor-specific CD4^+^ T cells. The decrease in lysosomal degradation of MHC-II in M-MDSCs by targeting the membrane-associated RING-CH1 (MARCH1) E3 ubiquitin ligase indirectly attenuates the suppressive function of M-MDSCs, resulting in markedly decreased tumor volume and a robust anti-tumor immunity [77]. Collectively, TME reshapes the differentiation of monocytes into M-MDSC at transcriptional and post-transcriptional levels.

Accumulation of M-MDSCs in fibrotic livers is associated with reduced tumor-infiltrating lymphocytes (TILs) and increases tumorigenicity in both mouse models and human cancer patients [78]. In human HCC, livers are markedly enriched with M-MDSC, with its surrogate marker CD33 significantly associated with aggressive tumor phenotypes and poor survival rates [79]. Activated human hepatic stellate cells (HSCs) induce monocyte-intrinsic p38 mitogen-activated protein kinase (p38 MAPK) signaling to trigger enhancer reprogramming for M-MDSC development and immunosuppression. Treatment with p38 MAPK inhibitor can abrogate HSC-M-MDSC crosstalk to prevent HCC growth [78].

## 4. The Paradoxical Roles of Monocytes in Tumor Metastasis

Metastasis is termed the spread and colonization of tumor cells in distant tissues. Classical “inflammatory” monocytes promote metastatic recurrence when systemic or local inflammation escalates under therapeutic interventions for primary tumors [76,77]. In breast cancer, classical monocytes are recruited to tumor sites, where they convert to macrophages and promote breast cancer metastasis [80]. In addition, a subset of Ym1^+^ Ly6C^hi^ monocytes that promote tissue repair [81] is also involved in this process. Depletion of Ym1^+^Ly6C^hi^ monocytes inhibits inflammation-induced metastasis, whilst re-introducing Ym1^+^Ly6C^hi^ cells into naïve mice promotes lung metastasis. The highly expressed matrix metalloproteinase-9 (MMP-9) and CXCR4 in Ym1^+^Ly6C^hi^ monocytes contribute to lung metastasis [28]. Moreover, monocytes are recently recognized to increase the invasive potential of head and neck cancer stem cells via promoting CD44-VCAM-1mmp binding and inducing Ezrin/PI3K activation [82,83]. Blocking monocyte-cancer interaction reverses the invasive phenotype [84]. 

Inflammation-induced monocyte-derived macrophages (CD11b^+^CD11c^+^ macrophages) increase the efficiency of early metastatic colonization in murine models [85]. Hepatocyte growth factor (HGF), secreted by CD11b^+^CD11c^+^ macrophages, augments tumor cell survival under stress conditions in vitro. It is further validated by the finding that blocking HGF signaling abolishes inflammation-induced early micro-metastatic lesion formation in vivo. These findings indicate that HGF-producing CD11b^+^CD11c^+^ macrophages contribute to a pre-metastatic niche and facilitate tumor cell survival within early micro metastases [85]. M-MDSCs perform indispensable roles in cancer metastasis by orchestrating complex crosstalk between breast cancer cells and surrounding stroma cells [86].

On the other hand, certain monocyte subsets may directly inhibit cancer metastasis. When pancreatic ductal adenocarcinoma (PDAC) cells are co-cultured with undifferentiated monocytes, invadopodia formation is significantly suppressed, which can be especially ascribed to tissue inhibitor of metalloproteinase-2 (TIMP2) secreted by monocytes. Thus, activation of TIMP2 expressing monocytes in primary tumors could pose a potential therapeutic opportunity for suppressing cell invasion and metastasis in PDAC [87]. IFN-γ-producing immune effector monocytes recruited to the lung by orthotopic tumors systemically upregulate the TMEM173/STING pathway in neutrophils and enhance their killing capacity. In this way, immune effector monocytes and tumoricidal neutrophils target disseminated tumor cells in the lungs, preventing metastatic outgrowth [88]. In addition, IFN-γ-induced intermediate monocytes (called IFN-IMos) inhibit cancer lung metastasis by promoting NK cell expansion through an IL-27-dependent pathway [89]. Forkhead box O1 (FOXO1) and nuclear receptor subfamily 4 group A member 1 (NR4A1) are discovered to regulate IFN-γ-driven monocyte differentiation and anti-metastatic activity of IFN-IMos. This finding renders two potential targets for improving IFN-γ treatment effect on cancer metastasis [89]. Different from classical monocytes, the non-classical “patrolling” monocytes are enriched in the micro-vasculature of the lung and reduce tumor metastasis to the lung in multiple murine metastatic tumor models [90] (Figure 3). These observations indicate the high plasticity of monocytes in the tumor metastasis.

## 5. Therapeutic Strategies Targeting Tumor-Related Monocytes

### 5.1. Targeting Monocyte Differentiation and Reprogramming

As myeloid precursor cells, monocytes replenish TAMs and M-MDSCs to support the establishment of suppressive TME, inhibition of CSF1-CSF1R signaling could decrease the numbers of TAMs and M-MDSCs along with other myeloid cells and improve T cell response in several tumor models [91,92]. GW2580, a CSF1R kinase inhibitor, reduces M2 macrophage infiltration and significantly decreases the volume of ascites in advanced ovarian cancer patients [93]. In addition, CSF1R antibodies (such as Emactuzumab) are also developed to block the CSF1-CSF1R pathway and showed efficacy on TAMs reduction in diffuse-type giant cancer cells [94]. Retinoic acid, a metabolite of vitamin A1 produced by murine sarcoma tumor cells, selectively suppresses the DC-promoting transcription factor IRF4 and drives intra-tumoral monocyte differentiation toward TAMs and away from DCs [18]. Genetic inhibition of retinoic acid production in tumor cells or pharmacologic inhibition of retinoic acid signaling within TME increases monocyte-derived DCs, enhances T cell-dependent anti-tumor immunity, and synergizes with immune checkpoint blockade [18]. Neurotransmitter gamma-aminobutyric acid (GABA) derived from B cells promotes monocyte differentiation into anti-inflammatory macrophages that secrete IL-10 and inhibit CD8^+^ T cell function, which can be reversed by B cell-specific inactivation of the GABA-generating enzyme GAD67 [95]. Pancreatic cancer cell-derived sialic acid dictates monocyte to macrophage differentiation via signaling through the Siglec receptors Siglec-7 and Siglec-9, highlighting a critical role for sialylated glycans in controlling immune suppression and providing potential targets for cancer immunotherapy in PDAC [96].

Given that circulating monocytes are functionally heterogeneous [97], therapeutic strategies for reprogramming monocytes from a tumor-supporting phenotype towards a tumoricidal phenotype are of great interest. Artesunate treatment induces an increase in inflammatory monocytes with HLA-DR high expression and MCP1/IL-1β release. Additionally, these artesunate-programmed monocytes acquire the ability to kill leukemic cells [98]. TAMs have been ascertained as obstacles to chemotherapy and radiotherapy [99], and most of them are differentiated from monocytes recruited to tumor sites [100]. TAMs can be polarized into protective M1 or tumorigenic M2 types in accordance with their surface markers and immune functions [101]. Reprogramming TAMs represents an attractive immunotherapeutic strategy in cancer treatments [102]. Strategies, including targeting the CSF1-CSF1R pathway, CD44 pathway, TLR receptor, CD206 inhibition, and STAT inhibition, have been described in detail in this review [101]. Moreover, strategies to prevent monocyte-derived macrophages from polarizing into pro-tumoral M2 macrophages have also been advanced. Caspase activation maintains IL-4-induced monocyte-derived macrophages polarization. Emricasan, a pan-caspase inhibitor, could prevent the generation and the anti-inflammatory polarization of monocyte-derived macrophages ex vivo [103]. A low dose of type I IFN could effectively reprogram human monocyte-derived macrophages to CD169^+^ macrophages, which exhibit significantly enhanced phagocytotic and CD8^+^ T cell-activating capacities [104].

### 5.2. Targeting Monocyte Recruitment and Adhesion to Tumor Sites

Cancer cells secrete cytokines and chemokines that draw circulating monocytes from blood into neoplastic lesions and contribute to the differentiation of infiltrated monocytes [105]. For example, M2-like TAMs, which increase motility, invasion, and matrix spreading of chondrosarcomas (CHS) cells, are derived from monocytes induced by soluble molecules of CHS cells. RI-3, a urokinase receptor (uPAR)/formyl peptide receptor type 1(FPR1) inhibitor, successfully prevents both recruitment and infiltration of monocytes into tumor tissues in nude mice, reducing the number of TAMs that infiltrate CHS tumors [106]. RI-3 keeps FPR1 anchored to the cell membrane, making it unable to internalize and activate uPAR-triggered differentiation from monocytes to macrophages, FPR1-mediated monocytes chemotaxis [107].

The recruitment of monocytes to tumor sites is mediated primarily via CCL2-CCR2 chemotactic axis [108]. Thus, the disruption of this axis represents an attractive therapeutic target for anti-cancer treatment. Losartan, a type I angiotensin II receptor (AT1R) antagonist, inhibits CD11b^+^Ly6C^+^ monocytes recruitment into the lung by blocking CCR2 signaling, which finally suppresses lung metastasis [109]. This effect also suggests that other AT1R blocker drugs could be potentially repurposed for their use in cancer immunotherapy. Furthermore, Losartan, in combination with kinase inhibitor toceranib, exerts significant biological activity in dogs with metastatic osteosarcoma, supporting the evaluation of this drug combination in patients with pediatric osteosarcoma [110]. Another work in a CT26 murine colon carcinoma model also suggests that blockade of CCL2 (or macrophage chemoattractant protein-1, MCP-1) is sufficient to reduce circulating monocytes-derived TAMs in TME and has the ability to modestly alter tumor growth to treatment [111]. 

Although the majority of monocytes recruited to TME are converted to TAMs or M-MDSCs with immunosuppressive functions, monocytes are also shown to exert anti-tumor effects. Disordered cell proliferation, altered metabolism, and aberrant blood vessels can lead to a hypoxic microenvironment in human solid tumors [112]. Tumor hypoxia induces stabilization of the transcription factor hypoxia-inducible factor 1-alpha (HIF-1A), which drives transcriptional responses in both immune cells and cancer cells, thus influencing tumor development [113]. In human lung adenocarcinoma (LUAD) samples, HIF-1α-induced microRNA-210-3p (mir-210-3p) directly caps the 3′ untranslated region (UTR) of CCL2 mRNA from being translated. In contrast, mir-210-3p inhibition promotes monocyte recruitment and skews monocytes towards an antitumor M1 phenotype, thus restoring tumor regression [114]. These results collectively suggest that increased infiltrating monocytes might perform a tumoricidal role against LUAD solid tumors. Therefore, the plasticity of recruited monocytes needs to be further investigated.

### 5.3. Monocytes as Carriers to Deliver Antigens and Drugs

Studies show that DC vaccines stimulate cytotoxic T lymphocytes (CTLs) mostly through transferring antigens (Ags) to endogenous DCs [115] rather than their antigen-presenting cell (APC) activities [116]. Such Ag transfer functions have long been described for monocytes and monocyte-derived cells [117], implying that undifferentiated monocytes could function well as a vaccine modality and need not be differentiated to DCs to be effective [118]. Intravenous injection of undifferentiated monocytes posts Ag loading displays anti-tumor activity superior to DC vaccines in several cancer models, including aggressive intracranial glioblastoma [118,119]. Ag-loaded monocytes induce robust CTL responses via Ag transfer to splenic CD8^+^ DCs independently of monocyte APC activity. Additionally, indeed, an efficient gap junction-mediated Ag transfer pathway between monocytes and CD8^+^ DCs exists. This suggests that administration of tumor Ag-loaded undifferentiated monocytes may serve as a simple and efficacious immunotherapy for the treatment of human cancers [118]. 

Monocytes are potential candidates for the delivery of therapeutic agents to TME because of their tumor-accumulating nature [120,121]. Monocytes are capable of carrying nanoparticles encapsulating anti-tumor drugs and dropping off the cargo at tumor sites. With superior efficiency than free nanoparticles, bio-distribution analysis confirmed that tumors are the most reached among peripheral tissues [122]. Conjugated polymer nanoparticles (CPNs)-loaded monocytes can efficiently deliver CPNs into glioblastoma for improved photodynamic therapy [123]. However, such ex vivo monocyte preparation processes are labor-intensive and time-consuming [124]. To achieve in vivo loading of monocytes/macrophages with therapeutics, their natural phagocytic behavior to take up foreign or waste materials can be exploited [125]. Apoptotic body (AB) can be used as a carrier for delivering nanomedicine, as they can be phagocytosed by inflammatory Ly6C^+^ monocytes, which then actively infiltrate into the tumor center [126]. Furthermore, with the aid of a cell-mediated delivery system, monocyte-based treatment allows for not only retarding the growth of primary tumors but also maintain a potent immunity to prevent tumors from metastasis and recurrence [124].

### 5.4. Additional Therapeutic Strategies

Since monocytes also contribute to cancer deterioration by influencing the functions of other immune cells, targeting the crosstalk between monocytes and other cells is likely to be an effective therapeutic strategy. Monocytes and other myeloid cells express the NOX2 isoform of NADPH oxidase, which generates reactive oxygen species (ROS) [127] regulated by PI3K [128,129]. NOX2-derived ROS may be released extracellularly to suppress or control the function and viability of adjacent NK cells [130] and T cells [131]. Conversely, the phosphatidylinositol-4,5-bisphosphate-3 kinase-d (PI3Kd) inhibitor idelalisib inhibits the formation of NOX2-derived ROS from human monocytes and rescues NK cells from NOX2/ROS-dependent cell death. This finding is further confirmed by systemic treatment with idelalisib, which reduces the formation of lung metastases from intravenously injected melanoma cells but does not affect metastasis in Nox2^−/−^ mice or NK cell–deficient mice [132]. 

Interestingly, there is abundant evidence that microbiota affects the immune response to cancer [133], including Involvement in monocyte-mediated crosstalk with other immune cells [134]. One study uncovers that a microbiota-derived stimulator of interferon genes (STING) agonists induces IFN-I production by intra-tumoral monocytes to regulate macrophage polarization and NK cell-DC crosstalk, thus improving the efficacy of immune checkpoint blockade (ICB). It is also observed in individuals with melanoma treated with ICB that intra-tumoral IFN-I and immune compositional differences between responder and non-responder individuals can be metabolically synchronized by fecal microbiota transplantation [134]. This observation indicates that transfusion of the STING agonist-stimulated monocytes back into cancer patients might elevate intra-tumoral immune responses through type I IFN.

## 6. Conclusions

Monocytes perform many important roles during tumorigenesis. Growing evidence shows that phenotypic changes of peripheral blood monocytes and the uniquely expressed molecules can be used as signature markers for diagnosis, treatment, and prognosis. Due to the accessibility and clinical relevance of monocytes, it provides a fast and practical tool for being broadly used in clinical diagnosis. Monocyte reprogramming induced by TME performs an important role in the progress of tumors. Transcriptomics of circulating monocytes and emerging single-cell sequencing data have shown that different cancer types induce different genetic signatures. The interaction among immune cells, or between monocytes and tumor cells in TME, needs further explored. Monocytes are able to infiltrate into TME to exert pro-tumoral activity, while on the other hand, they may also suppress oncogenesis. The reason monocytes can develop in the opposite phenotypes remains to be clarified. Thanks to high-throughput technologies, new monocyte populations, and monocyte-derived cells have been discovered. Whether these cell subsets are universal in different cancers and whether the functions are unique warrant further research and exploitation. In addition, the short-term and long-term effects of infiltrating monocytes in TME and peripheral blood monocycles on different cancer treatment modalities are not fully understood. Treatment-induced monocyte changes and their roles in therapeutic resistance and disease progression are another relevant area of research in the future. The advances in our knowledge of monocyte development, response, and reprogramming, particularly during cancer evolution and metastatic spread, will pave the way for the development of specific and efficacious therapeutic strategies for cancer.

## Figures and Tables

**Figure 1 cells-12-01673-f001:**
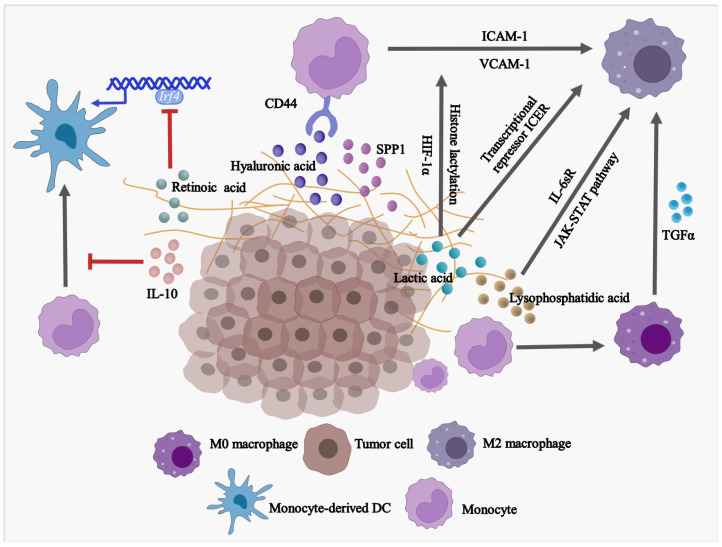
Factors in TME are driving the differentiation of monocytes into pro-tumoral macrophages while blocking the development of anti-tumor DCs. SPP1, secreted phosphoprotein 1/osteopontin; IL-10, Interleukin 10; TGFα, Transforming growth factor alpha; IRF4, Interferon regulatory factor 4; USP12, Ubiquitin specific peptidase 12; ICAM-1, Intercellular adhesion molecule 1; VCAM-1, Vascular cell adhesion molecule 1; HIF-1α, Hypoxia-inducible factor 1α; IL-6sR, Soluble interleukin 6 receptor.

**Figure 2 cells-12-01673-f002:**
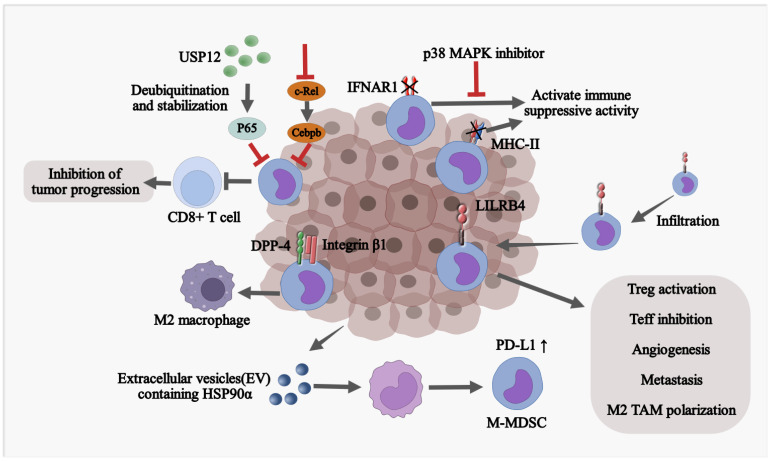
M-MDSCs acquire immunosuppressive phenotypes in TME through multiple mechanisms. CEBPB, CCAAT enhancer binding protein beta; IFNAR1, interferon alpha, and beta receptor subunit 1; LILRB4, leukocyte immunoglobulin-like receptor B4; DPP-4, dipeptidyl peptidase 4; PD-L1, programmed death-ligand 1; HSP90α, heat shock protein 90 alpha; ICAM-1, intercellular adhesion molecule 1; VCAM-1, vascular cell adhesion molecule 1.

**Figure 3 cells-12-01673-f003:**
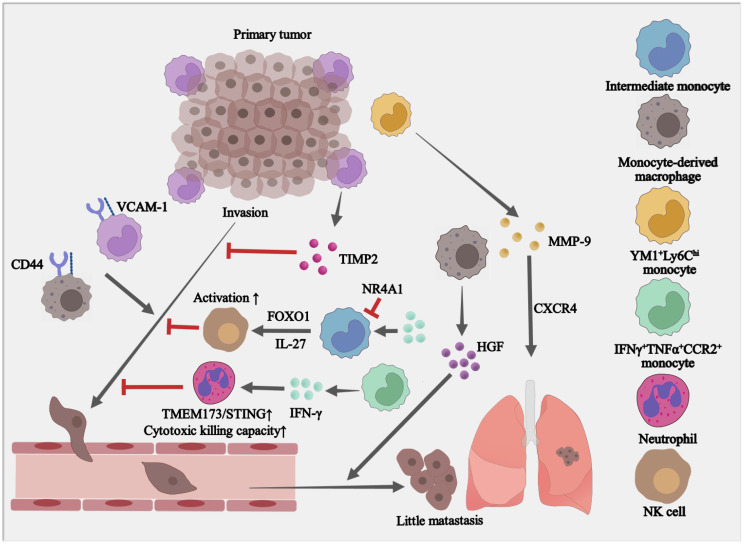
Monocytes affect the invasion and metastasis of tumor cells. VCAM-1, Vascular cell adhesion molecule 1; TIMP2, TIMP metallopeptidase inhibitor 2; IFN-γ, Interferon gamma; TMEM173/STING, Stimulator of interferon response CGAMP interactor 1; HGF, Hepatocyte growth factor; MMP-9, Matrix metallopeptidase 9; CCR2, C-C motif chemokine receptor 2; TNF-α, Tumor necrosis factor alpha. FOXO1, Forkhead box O1; NR4A1, Nuclear receptor subfamily 4 group A member 1.

**Table 1 cells-12-01673-t001:** Pro-tumoral and anti-tumoral functions of monocyte subsets and monocyte-derived cells in human cancer.

Subset	Cellular Origin		Function	Methods	Cancer Type
CD66b^+^CD14^+^CD33^hi^CD16^−/+^HLA-DR^+/hi^ monocytes	CD33^hi^CD14^+^ monocytes	Anti-tumoral	Display high phagocytic activity, matrix adhesion, and migration, and provide co-stimulation for T cell proliferation and interferon-γ (IFN-γ) secretion.	RNA-seq and flow cytometry	Breast cancer and colorectal cancer [43]
CXCL9^+^CXCL10^+^ CCL5^+^MHCII^+^ CD40^+^STAT1^+^ macrophages	-	Anti-tumoral	Secrete CXCL9 to facilitate recruitment of protective T cells.	scRNA-seq	Lung cancer [44,45]
CSFR1^+^CCR2^−^CD68^+^ CD163^+^SIGLEC1^−^ macrophages; CSFR1^+^CCR2^−^CD68^+^CD163^+^SIGLEC1^+^ macrophages; CSFR1^+^CCR2^−^CD68^+^CD163^−^SIGLEC1^+^ macrophages	CD14^++^CD16^−^CCR2^+^ classical monocytes	Pro-tumoral	Engage in a tumor cell-TAM auto-stimulatory loop, increase tumor cell motility, and increase monocyte infiltration into the tumor site to generate more pro-tumoral TAMs.	RNA-seq	Breast cancer [8]
TREM2^+^FOLR2^+^CD163^+^ macrophages	S100A8^+^ monocytes	Pro-tumoral	Recruit suppressive regulatory T cells (Treg) and MDSCs to facilitate immunosuppressive microenvironment.	scRNA-seq	Hepatocellular carcinoma [46]
CD14^+^CD16^+^(FCGR3A) CD81^+^ITGAX^+^CSF1R^+^ monocytes/macrophages	-	Pro-tumoral	Secrete specific profibrotic, pro-metastatic growth factors involved ECM deposition and remodeling.	scRNA-seq	Small cell lung cancer [47]
CD11b^+^CCR2^+^IL-1β^hi^Arg1^−^ M-MDSCs	CD11b^+^CCR2^+^ monocytes	Pro-tumoral	Promote tumor growth, suppress T cell function, and maintain suppressive TME.	scRNA-seq	Melanoma [42,48]
CD84^+^CD11b^+^/CD14^+^ M-MDSCs	PBMC	Pro-tumoral	Exhibit T cell suppression and increase ROS production.	scRNA-seq	Breast cancer [49]
CD14^+^HLA-DR^lo/−^ monocytes/MDSCs	CD14^+^HLA-DR^lo/−^ monocytes	Pro-tumoral	Inhibit T cell responses.	Flow cytometry	Epithelial ovarian cancer [50]

Abbreviations: ECM, extracellular matrix; ROS, reactive oxygen species; STAT1, signal transducer and activator of transcription 1; SIGLEC1, sialic acid binding Ig, such as lectin 1; TREM2, triggering receptor expressed on myeloid cells 2; FOLR2, folate receptor beta; ITGAX, integrin subunit alpha X; CSF1R, colony stimulating factor 1 receptor; PBMC, peripheral blood mononuclear cell.

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
