# Peer review of "Monocytes in Tumorigenesis and Tumor Immunotherapy"

_cells, 2023, doi:10.3390/cells12131673_

Round 1

Reviewer 1 Report

There is a lot excellent review on tumor associated macrophages that support tumor progression and make the important different in shaping tumor microenvironment. The authors produced a study in which they recapitulate most of the information which is well known and discussed. This review is a compilation of the well-known information but the focus on monocytes and their fates in TME makes it interesting. They described in details features of distinct subsets of monocytes and their functions . The only weakness is the lack of discussion of sc-omics data and insigths from animal models that lately in some tumors provided novel information. The parts devoted to differentiation of monocytes in TME and factors controlling those events nicely summarized the information in different cancer which are the factors and mechanisms. Sections on therapeutic approaches targeting or using monocytes as delivery tools are interesting and well discussed. The review is well written and the leading mechanisms are well illustrated.

Minor comments:

1)     While presenting all the published data on distinct subsets of monocytes and their functions in a table is useful one would appreciate more deep discussion what all the data means. Are the events in different tumors similar or not? Are signals alike or tumor specific?

2)     Expanding discussion of monocytic phenotypes and information which are detected in blood and which in tissue would be useful. The Table 1 shows antitumor phenotypes of cells in breast cancers, but

3)     Some comments are imprecise and misleading: Monocytes and their derivatives, such as myeloid derived suppressor cells (MDSCs) and tumor‐associated macrophages”. MDSCs and TAMs are not derivatives, those are monocytes derived, reprogrammed cells.

4)     The author statement “Inhibition of CSF1‐CSF1R signaling disturbs the monocyte differentiation into TAMs” is misleading as the CSF1R inhibition blocks survival of all myeloid cells (not only monocytes) thus reducing their numbers everywhere including in TME.

The review is well written, but in some parts language needs some improvement i.e. Tumorigenesis can be resulted from mutated genes; single‐cell biotechnologies should be single‐cell technologies

The review is well written, but in some parts language needs some improvement i.e. Tumorigenesis can be resulted from mutated genes; single‐cell biotechnologies should be single‐cell technologies

Author Response

We thank both reviewers for their constructive and thoughtful comments. In this revised version, we have addressed all their concerns/comments thoroughly. As a result, the quality of this review article has been significantly improved. The file below is the summary of the changes made in response to the reviewers’ comments.

Reviewer 2 Report

Review of ”Monocytes in tumorigenesis and tumor immunotherapy” by Chen et al.

This is an interesting and well-written review. The review covers many aspects of monocyte biology in relation to cancer, also highlighting interesting new angles.

Points to consider:

1    - The manuscript would benefit from a description of the immune and phagocytosis suppressing SIRP-alfa receptor expressed on different monocytes, which interacts with CD47 expressed by many types of cancer. Blocking anti-CD47 Abs are promising novel therapeutics against cancer.

2      - Monocyte plasticity between M1 – M2 stages is mentioned in the manuscript, however changes in intracellular metabolism seems to be of great importance for this plasticity. Since metabolism (and tumor metabolism) can be changed therapeutically, an incorporation of these points would strengthen the manuscript and make it more up to date.

Author Response

(The authors gave the same response as above.)
